# CAN I TRUST YOU MORE?
# MODEL-AGNOSTIC HIERARCHICAL EXPLANATIONS

## ABSTRACT

Interactions such as double negation in sentences and scene interactions in images are common forms of complex dependencies captured by state-of-the-art machine learning models. We propose Mahé, a novel approach to provide **M**odel-**a**gnostic **h**ierarchical **é**xplanations of how powerful machine learning models, such as deep neural networks, capture these interactions as either dependent on or free of the context of data instances. Specifically, Mahé provides context-dependent explanations by a novel local interpretation algorithm that effectively captures any-order interactions, and obtains context-free explanations through generalizing context-dependent interactions to explain global behaviors. Experimental results show that Mahé obtains improved local interaction interpretations over state-of-the-art methods and successfully provides explanations of interactions that are context-free.

## 1 INTRODUCTION

State-of-the-art machine learning models, such as deep neural networks, are exceptional at modeling complex dependencies in structured data, such as text (Vaswani et al., 2017; Tai et al., 2015), images (He et al., 2016; Huang et al., 2017), and DNA sequences (Alipanahi et al., 2015; Zeng et al., 2016). However, there has been no clear explanation on what type of dependencies are captured in the black-box models that perform so well (Ribeiro et al., 2018; Murdoch et al., 2018).

In this paper, we make one of the first attempts at solving this important problem through interpreting two forms of structures, i.e., context-dependent representations and context-free representations. A context-dependent representation is the one in which a model's prediction depends specifically on a data instance level (such as a sentence or an image). In order to illustrate the concept, we consider an example in image analysis. A yellow round-shape object can be identified as the sun or the moon given its context, either bright blue sky or dark night. A context-free representation is one where the representation behaves similarly independent of instances (i.e., global behaviors). In a hypothetical task of classifying sentiment in sentences, each sentence carries very different meaning, but when "not" and "bad" depend on each other, their sentiment contribution is almost always positive - i.e., the structure is context-free.

To investigate context-dependent and context-free structure, we lend to existing definitions in interpretable machine learning (Ribeiro et al., 2016; Kim et al., 2018). A context-dependent interpretation is a local interpretation of the dependencies at or within the vicinity of a single data instance. Conversely, a context-free interpretation is a *global* interpretation of how those dependencies behave in a model irrespective of data instances. In this work, we study a key form of dependency: an *interaction* relationship between the prediction and input features. Interactions can describe arbitrarily complex relationships between these variables and are commonly captured by state-of-the-art models like deep neural networks (Tsang et al., 2018; Murdoch et al., 2018). Interactions which are context-dependent or context-free are therefore local or global interactions, respectively.

We propose Mahé, a framework for explaining the context-dependent and context-free structures of any complex prediction model, with a focus on explaining neural networks. The context-dependent explanations are built based on recent work on local intepretations (such as (Ribeiro et al., 2016; Murdoch et al., 2018; Singh et al., 2018)). Specifically, Mahé takes as input a model to explain and a data instance, and returns a hierarchical explanation, a format proposed by Singh et al. (2018) to show local group-variable relationships used in predictions (Figure 1). To provide context-free

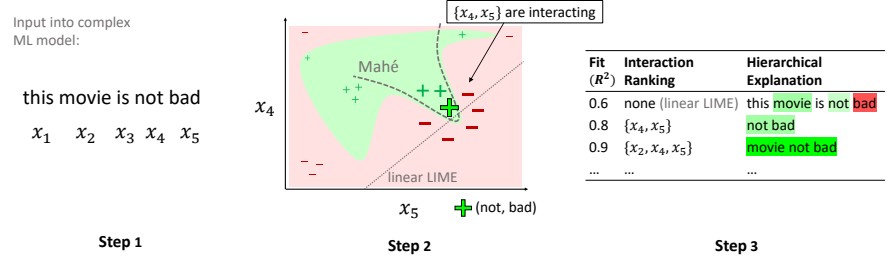

Figure 1: An overview of the steps used to obtain a context-dependent hierarchical explanation. Step 1 inputs a data instance of interest (e.g. a sentence) into a complex model, in this case a classifier. Step 2 locally perturbs the data instance and obtains their predictions results from the model. Instead of only fitting a linear model as linear LIME does to the perturbed samples and outputs, Mahé fits a neural network to them to learn the highly nonlinear decision boundary used to classify the instance. The nonlinearity indicates that there should be an interaction between variables, and an interpretation of the neural network is used to extract the interactions (Tsang et al., 2018). Attribution scores of those interactions can then be shown for the data instance, as displayed in Step 3.

explanations, Mahé generalizes those context-dependent interactions with consistent behavior in a model and determines whether a local representation in the model is responsible for the global behavior. In this case, Mahé takes as input a model and representative data corresponding to an interaction of interest and returns whether or not that interaction is context-free. We conduct experiments on both synthetic datasets and real-world application datasets, which shows that Mahé's context-dependent explanations can significantly outperform state-of-the-art methods for local interaction interpretation, and Mahé is capable of successfully finding context-free explanations of interactions. In addition, we identify promising cases where the methodology for context-free explanations can successfully edit models. Our contributions are as follows: 1) Mahé achieves the task of improved context-dependent explanations based on interaction detection and fitting performance and model-agnostic generality, compared to state-of-the-art methods for local interaction interpretation, 2) Mahé is the first to provide context-free explanations of interactions in deep learning models, and 3) Mahé provides a promising direction for modifying context-free interactions in deep learning models without significant performance degradation.

## 2 RELATED WORKS

**Attribution Interpretability**: A common form of interpretation is feature attribution, which is concerned with how features of a data instance contribute to a model output. Within this category, there are two distinct approaches: additive and sensitivity attribution. Additive attribution interprets how much each feature contributes to the model output when these contributions are summed. In contrast, sensitivity attribution interprets how sensitive a model output is to changes in features. Examples of additive attribution techniques include LIME (Ribeiro et al., 2016) and CD (Murdoch et al., 2018). Examples of sensitivity attribution methods include Integrated Gradients (Sundararajan et al., 2017), DeepLIFT (Shrikumar et al., 2017), and SmoothGrad (Smilkov et al., 2017). Unlike previous approaches, Mahé provides additive attribution interpretations that consist of *non-additive groups* of variables (interactions) in addition to the normal additive contributions of each variable.

**Interaction Interpretability**: An interaction in its generic form is a non-additive effect between features on an outcome variable. Only until recently has there been development in interpreting non-additive interactions despite often being learned in complex machine learning models. The difficulty interpreting non-additive interactions stems from their lack of exact functional identity compared to, for example, a multiplicative interaction. Methods that exist to interpret non-additive interactions are NID (Tsang et al., 2018) and Additive Groves (Sorokina et al., 2008). In contrast, many more methods exist to interpret specific interactions, namely multiplicative ones. Notable methods include CD (Murdoch et al., 2018), Tree-Shap (Lundberg & Lee, 2017), and GLMs with multiplicative interactions (Purushotham et al., 2014). Unlike previous methods, our approach provides local interpretations of the more challenging *non-additive interaction*.

**Locally Interpretable Model-Agnostic Explanations (LIME)**: LIME (Ribeiro et al., 2016) is a very popular type of model interpretation. Its popularity comes from additive attribution interpretations to explain the output of *any* prediction model. The original and most popular version of LIME uses a linear model to approximate model predictions in the local vicinity of a data instance. Since its introduction, variants of LIME have been proposed, for example Anchors (Ribeiro et al., 2018)

and LIME-SUP (Hu et al., 2018). While Anchors generates a form of context-free explanation, its method of selecting fully representative features for a prediction does not consider interactions. For example, Anchors assumes that (not, bad) "virtually guarentees" a sentiment prediction to be positive, whereas in Mahé this is not necessarily true; only their *interaction* is positive (See Table 6 for an example). LIME-SUP touches upon interactions but does not study their interpretation.

## 3 INTERACTION EXPLANATIONS

Let $f(\cdot)$ be a target function (model) of interest, e.g. a classifier, and $\phi(\cdot)$ be a local approximation of $f$ and is interpretable in contrast to $f$. A common choice for $\phi$ is a linear model, which is interpretable in each linear term. Namely, for an data instance $\mathbf{x} \in \mathbb{R}^p$, weights $\mathbf{w} \in \mathbb{R}^p$ and bias $b$, interpretations are given by $w_i x_i$, known as *additive attributions* (Lundberg & Lee, 2017) from

$$\phi(\mathbf{x}) = \sum_{i=1}^{p} w_i x_i + b. \tag{1}$$

Given a set of $n$ data points $\{\mathbf{x}^{(i)}\}_{i=1}^{n}$ that are infinitesimally close or local to $\mathbf{x}$, a linear approximation of $\mathcal{D} = \{(\mathbf{x}^{(1)}, f(\mathbf{x}^{(1)})), \ldots, (\mathbf{x}^{(n)}, f(\mathbf{x}^{(n)}))\}$ will accurately fit to the functional surface of $f$ at the data instance, such that $\phi(\mathbf{x}) = f(\mathbf{x})$. Because it is possible in such scenarios that $\phi(\mathbf{x}) = f(\mathbf{x}) \approx b$, there must be some nonzero distances between $\mathbf{x}$ and $\mathbf{x}^{(i)}$ to obtain informative attribution scores. LIME, as it was originally proposed, uses a linear approximation as above where samples are generated in a nonzero local vicinity of $\mathbf{x}$ (Ribeiro et al., 2016). The drawback of linear LIME is that there is often an error $\epsilon = |f(\mathbf{x}) - \phi(\mathbf{x})| > 0$.

For complex models $f$, the functional surface at $\mathbf{x}$ can be nonlinear. Because $\mathcal{D}$ consists of $\mathbf{x}^{(i)}$ with distance $d > 0$ from $\mathbf{x}$, a closer fit to $f(\mathbf{x})$ in its nonlinear vicinity, i.e. $\{f(\mathbf{x}^{(i)})\}_{i=1}^{n}$, can be achieved with the following generalization of Eq. 1:

$$\phi(\mathbf{x}) = \sum_{i=1}^{p} g_i(x_i) + b, \tag{2}$$

where $g_i(\cdot)$ can be any function, for example one that is arbitrarily nonlinear. This function is called a generalized additive model (GAM) (Hastie & Tibshirani, 1990), and now attribution scores can be given by $g_i(x_i)$ for each feature $i$. For the purposes of interpreting individual feature attribution, the GAM may be enough. However, if we would like broader explanations, we can also obtain non-additive attributions or interactions between variables (Lou et al., 2013), which can provide an even better fit to the complex local vicinity. Expanding Eq. 2 with interactions yields:

$$\phi_K(\mathbf{x}) = \sum_{i=1}^{p} g_i(x_i) + \sum_{i=1}^{K} g_i'(\mathbf{x}_{\mathcal{I}}) + b, \tag{3}$$

where $g_i'(\cdot)$ can again be any function, $\mathbf{x}_{\mathcal{I}} \in \mathbb{R}^{|\mathcal{I}|}$ are interacting variables corresponding to the variable indices $\mathcal{I}$, and $\{\mathcal{I}_i\}_{i=1}^{K}$ is a set of $K$ interactions. Attribution scores are now generated from both $g_i'$ and $g_i'$. In this paper, we learn $g_i$ and $g_i'$ using Multilayer Perceptrons (MLPs). $\phi$ or $\phi_K$ can be converted to classification by applying a sigmoid function.

Adding non-additive interactions, $\mathcal{I}$, that are truly present in the local vicinity increases the representational capacity of $\phi_K(\mathbf{x})$. $\mathcal{I}$ corresponds to non-additive interacting features if and only if $g'(\cdot)$ (Eq. 3) cannot be decomposed into a sum of $|\mathcal{I}|$ arbitrary subfunctions $\delta$, each not depending on a corresponding interacting variable (Tsang et al., 2018), i.e.

$$g'(\mathbf{x}_{\mathcal{I}}) \neq \sum_{i \in \mathcal{I}} \delta_i(\mathbf{x}_{\mathcal{I} \setminus \{i\}}).$$

## 4 Mahé FRAMEWORK

In this section, we introduce our Mahé framework, which can provide context-dependent and context-free explanations of interactions. To provide context-dependent explanations, we propose to use a two-step procedure that first identifies what variables interact locally, then learns a model of interactions (as Eq. 3) to provide a local interaction score at the data instance in question. The procedure of first detecting interactions then building non-additive models for them has been studied previously (Lou et al., 2013; Tsang et al., 2018); however, previous works have not focused on using the same non-additive models to provide local interaction attribution scores, which enable us to visualize interactions of any size as demonstrated later in §5.2.3.

### 4.1 CONTEXT-DEPENDENT EXPLANATIONS

**Local Interaction Detection**: To perform interaction detection on samples in the local vicinity of data instance $\mathbf{x}$, we first sample $n$ points in the $\epsilon$-neighborhood of $\mathbf{x}$ with a maximum neighborhood distance $\epsilon$ under a distance metric $d$. While the choice of $d$ depends on the feature type(s) of $\mathbf{x}$, we always set $\epsilon = \sigma$, i.e. one standard deviation from the mean of a Gaussian weighted sampling kernel. When all features are continuous, neighborhood points are sampled with mean $\mathbf{x} \in \mathbb{R}^p$ and $d = \ell_2$ to generate $\mathbf{x}^{(1)}, \ldots, \mathbf{x}^{(n)}$, $\mathbf{x}^{(i)} \sim \mathcal{N}(\mathbf{x}, \sigma^2 \mathbf{I})$, where $\mathcal{N}$ is a normal distribution truncated at $\epsilon$. When features are categorical, they are converted to one-hot binary representation. For $\mathbf{x}$ of binary features, we sample each point around $\mathbf{x}$ by first selecting a number of random features to flip (or perturb) from a uniform distribution between 0 and $\min(p, \epsilon')$. The max number of flips $\epsilon'$ is derived from $\epsilon$ for a distance metric that is usually cosine distance (Ribeiro et al., 2016). Distances between local samples and $\mathbf{x}$ are then weighted by a Gaussian kernel to become sample weights (e.g. the frequency each sample appears in the sampled dataset).[1] For context-dependent explanations, the exact choice of $\sigma$ depends on the stability and interaction orders of explanations. The interaction orders may become too large and uninformative because the local vicinity area covers too much complex representation from $f(\cdot)$. Thus we recommend tuning $\sigma$ to the task at hand.

Our framework is flexible to any interaction detection method that applies to the dataset $\mathcal{D} = \{(\mathbf{x}^{(1)}, f(\mathbf{x}^{(1)})), \ldots, (\mathbf{x}^{(n)}, f(\mathbf{x}^{(n)}))\}$. Since we seek to detect non-additive interactions, we use the neural interaction detection (NID) framework (Tsang et al., 2018), which interprets learned neural network weights to obtain interactions. To the best of our knowledge, this detection method is the only polynomial-time algorithm that accurately ranks any-order non-additive interactions after training one model, compared to alternative methods that must train an exponential number $O(2^p)$ of models. The basic idea of NID is to interpret an MLP's accurate representation of data to accurately identify the statistical interactions present in this data. Because MLPs learn interactions at nonlinear activation functions, NID performs feature interaction detection by tracing high-strength $\ell_1$-regularized weights from features to common hidden units. In particular, NID efficiently detects any-order interactions by first assuming each first layer hidden unit in a trained MLP captures at most one interaction, then NID greedily identifies these interactions and their strengths through a 2D traversal over the MLP's input weight matrix, $\mathbf{W} \in \mathbb{R}^{p \times h}$. The result is that instead of testing for interactions by training $O(2^p)$ models, now only $O(1)$ models and $O(ph)$ tests are needed.

In addition to its efficiency, applying NID to our framework Mahé has several advantages. One is the universal approximation capabilities of MLPs (Hornik, 1991), allowing them to approximate arbitrary interacting functions in the potentially complex local vicinity of $f(\mathbf{x})$. Another advantage is the independence of features in the sampled points of $\mathcal{D}$. Normally, interaction detection methods cannot identify high interaction strengths involving a feature that is correlated with others because interaction signals spread and weaken among correlated variables (Sorokina et al., 2008). Without facing correlations, NID can focus more on interpreting the data-generating function, the target model $f$. One disadvantage of our application of NID is the curse of dimensionality for MLPs when $p$ is large (e.g. $p > n$) (Theodoridis et al., 2008), which is oftentimes the case for images. In general, large input dimensions should be reduced as much as possible to avoid overfitting. For images, $p$ is normally reduced in model-agnostic explanation methods by using segmented aggregations of pixels called superpixels as features (Ribeiro et al., 2016; Lundberg & Lee, 2017; Ribeiro et al., 2018).

**Hierarchical Interaction Attributions**: Upon obtaining an interaction ranking from NID, GAMs with interactions (Eq. 3) can be learned for different top-$K$ interactions ranked by their strengths (Tsang et al., 2018). In the Mahé framework, there are $L + 1$ different levels of a *hierarchical explanation* which constitutes our context-dependent explanation, where $L$ is the number of levels with interaction explanations, and $K = L$ at the last level. When presenting the hierarchy such as Figure 1 Step 3, the first level shows the additive attributions of individual features from by a trained $\phi(\cdot)$ in Eqs. 1 or 2, such as the explanation from linear LIME. Subsequently, the parameters $\mathbf{w}$ of $\phi(\cdot; \mathbf{w}, b)$ are frozen before interaction models are added to construct $\phi_K(\cdot)$ in Eq. 3. The next levels of the hierarchy can be presented either as the interaction attribution of $g'_K(\cdot)$ as in Figure 1 or

---

[1] In cases where features are a mixture of continuous and one-hot categorical variables, a way of sampling points is to adapt the approach for binary features to handle the mixture of feature types (Ribeiro et al., 2016). The main difference now is that continuous features are drawn from a uniform distribution truncated at $\sigma$ and are standard scaled to have similar magnitudes as the binary features. Since continuous features are present, $d$ can be $\ell_2$ distance, then a Gaussian kernel can be applied to sample distances as before.

those of $\{g_i'(\cdot)\}_{i=1}^K$ (Eq. 3), where at each level $K$ is increased and $\{g_i'(\cdot)\}_{i=1}^K$, bias $b$ are retrained. The practice of training interaction models $g_i'$ on the residual of $\phi$ is used to prevent degeneracy of univariate functions in $\phi$ in the presence of any overlapping interaction functions (Lou et al., 2013). Since $\phi_K$ is trained at each hierarchical level on $\mathcal{D}$, the fit of each $\phi_K$ can also be explained via predictive performance, such as $R^2$ performance in Figure 1 Step 3. The stopping criteria for the number of hierarchical levels can depend on the predictive performance or user preference.

## 4.2 CONTEXT-FREE EXPLANATIONS

In order to provide context-free explanations, we propose determining whether the local interactions assumed to be context-dependent in §4.1 can generalize to explain global behavior in $f$. To this end, we first define ideal conditions for which a generic local explanation can generalize. For choosing distance metric $d$ and sampling points in the local vicinity of $\mathbf{x}$, please refer to §4.1 and our considerations for generalizing explanations at the end of this section.

**Definition 1** (Generalizing Local Explanations). *Let $f(\cdot)$ be the model output we wish to explain, and $\mathcal{X}_f$ be the data domain of $f$. Let a local explanation of $f$ at $\mathbf{x} \in \mathcal{X}_f$ be some explanation $E$ that is true for $f(\mathbf{x})$ and depends on samples $\mathbf{x}_\ell \in \mathcal{X}_f$ that are only in the local vicinity of $\mathbf{x}$, i.e. $d(\mathbf{x}, \mathbf{x}_\ell) \leq \epsilon$ provided a distance metric $d$ and distance $\epsilon \geq 0$. The local explanation $E$ is a global explanation if the following two conditions are met: 1) Explanation $E$ is true for $f$ at all data samples in $\mathcal{X}_f$, including samples outside the local vicinity of $\mathbf{x}$, i.e. all samples $\mathbf{x}_g \in \mathcal{X}_f$ satisfying $d(\mathbf{x}, \mathbf{x}_g) > \epsilon$. 2) There exists a sample $\mathbf{x}' \in \mathcal{X}_f$ and a local modification to $f(\mathbf{x}')$ (modifying $f(\mathbf{x}_\ell)$ in the vicinity $d(\mathbf{x}', \mathbf{x}_\ell) \leq \epsilon$) that changes $E$ for all samples in $\mathcal{X}_f$ while still meeting condition 1).*

For example, consider a simple linear regression model we wish to explain, $f(\mathbf{x}) = w_1 x_1 + w_2 x_2$. Let its local explanation be the feature attributions $w_1 x_1$ and $w_2 x_2$. This local explanation is a global explanation because 1) for all values of $x_1$ and $x_2$, the feature attributions are still $w_1 x_1$ and $w_2 x_2$, and 2) if any of the weights are changed, e.g. $w_1 \to w_1'$, the attribution explanation will change, but the feature attributions are still $w_1' x_1$ and $w_2 x_2$ for all values of $x_1$ and $x_2$.

Our context-free explanation of interaction $\mathcal{I}$ is: whenever local interaction $\mathcal{I}$ exists, its attribution will in general have the same polarity (or sign). Since it is impossible to empirically prove that a local explanation is true for all data instances globally (via Definition 1), this work is focused on providing *evidence* of context-free interactions. This evidence can be obtained by checking whether our explanation is consistent with the two conditions from Definition 1 for the interaction of interest $\mathcal{I}$: 1) For representative data instances in the domain of $f$, if local interaction $\mathcal{I}$ exists, does it always have the same attribution polarity? The representative data instances should be separated from each other at an average distance beyond $\epsilon$. 2) Can local interaction $\mathcal{I}$ at a single data instance $\tilde{\mathbf{x}}$ be used to negate $\mathcal{I}$'s attribution polarity for all representative data instances where $\mathcal{I}$ exists?

The advantage of checking the response of $f$ to local modification is determining if consistent explanations across data instances are more than just coincidence. This is especially important when only a limited number of data instances are available to test on. We propose to modify an interaction attribution of the model's output $f(\mathbf{x})$ at data instance $\mathbf{x}$ by utilizing a trained model $g_k'(\mathbf{x}_\mathcal{I})$ of interaction $\mathcal{I}_k$, where $1 \leq k \leq K$ (Eq. 3). Let $\tilde{g}_k'(\cdot)$ be a modified version of $g_k'(\cdot)$. We can then define a modified form of Eq. 3:

$$\tilde{\phi}_k(\mathbf{x}) = \phi(\mathbf{x}) + \tilde{g}_k'(\mathbf{x}_\mathcal{I}) + \sum_{i=1, i\neq k}^K g_i'(\mathbf{x}_\mathcal{I}). \quad (4)$$

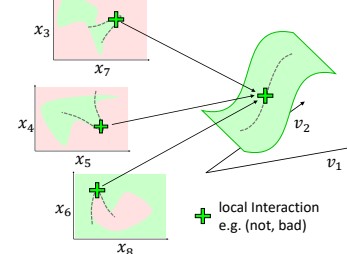

Without retraining $\tilde{\phi}_k(\cdot)$, we use $\tilde{\phi}_k$ and the same local vicinity $\{\mathbf{x}^{(i)}\}_{i=1}^n$ in $\mathcal{D}$ to generate a new dataset $\tilde{\mathcal{D}} = \{(\mathbf{x}^{(1)}, \tilde{\phi}_k(\mathbf{x}^{(1)})), \ldots, (\mathbf{x}^{(n)}, \tilde{\phi}_k(\mathbf{x}^{(n)}))\}$. Finally, we can modify the interaction attribution of $f(\mathbf{x})$ by fine-tuning $f(\cdot)$ on dataset $\tilde{\mathcal{D}}$. In this paper, we modify interactions by negating them: $\tilde{g}_k'(\cdot) = -c g_k'(\cdot)$, where $-c$ negates the interaction attribution with a specified magnitude $c$.

Figure 2: An illustration of the hypothesis that certain local interactions, which are similar (left), are represented at a common manifold (right) in a model.

How can modifying a local interaction affect interactions outside its local vicinity? This would suggest that the manifold hypothesis is true for $f(\cdot)$'s representations of these interactions (Figure 2). The manifold hypothesis states that similar data lie near a low-dimensional manifold in a high-dimensional space (Turk & Pentland, 1991; Lee et al., 2003; Cayton,

2005). Studies have suggested that the hypothesis applies to the data representations learned by neural networks (Rifai et al., 2011; Basri & Jacobs, 2016). The hypothesis is frequently used to visualize how deep networks represent data clusters (Maaten & Hinton, 2008; LeCun et al., 2015), and it has been applied to representations of interactions (Reed et al., 2014), but not for neural networks.

Part of our objective is to generalize our explanation as much as possible. In the case of language-related tasks, we additionally generalize based on our meaning of a local interaction and the distance metric we use, $d$. In this paper, local interactions for language tasks do not have word interactions fixed to specific positions; instead, these interactions are only defined by the words themselves (the interaction values) and their positional order. For example, the ("not", "bad") interaction would match in the sentences: "this is not bad" and "this does not seem that bad". For comparing texts and measuring vicinity sizes, we use edit distance (Levenshtein, 1966), which allows us to compare sentences with different word counts.[2] Although we define distance metrics for each domain (§5.1), we found that our results were not very sensitive to the exact choice of valid distance metric.

# 5 EXPERIMENTS

## 5.1 EXPERIMENTAL SETUP

We evaluate the effectiveness of `Mahé` first on synthetic data and then on four real-world datasets. To evaluate context-dependent explanations of `Mahé`, we first evaluate the accuracy of `Mahé` at local interaction detection and modeling on the outputs of complex base models trained on synthetic ground truth interactions. We compare `Mahé` to Shap-Tree (Lundberg et al., 2018), ACD-MLP (Singh et al., 2018), and ACD-LSTM (Murdoch et al., 2018; Singh et al., 2018), which are local interaction modeling baselines for the respective models they explain: XGBoost (Chen & Guestrin, 2016), multilayer perceptrons (MLP), and long short-term memory networks (LSTM) (Hochreiter & Schmidhuber, 1997). Synthetic datasets have $p = 10$ features (Table 2).

In all other experiments, we study `Mahé`'s explanations of state-of-the-art level models trained on real-world datasets. The state-of-the-art models are: 1) DNA-CNN, a 2-layer 1D convolutional neural network (CNN) trained on MYC-DNA binding data [3] (Mordelet et al., 2013; Yang et al., 2013; Alipanahi et al., 2015; Zeng et al., 2016; Wang et al., 2018), 2) Sentiment-LSTM, a 2-layer bi-directional LSTM trained on the Stanford Sentiment Treebank (SST) (Socher et al., 2013; Tai et al., 2015), 3) ResNet152, an image classifier pretrained on ImageNet '14 (Russakovsky et al., 2015; He et al., 2016), and 4) Transformer, a machine translation model pretrained on WMT-14 En→ Fr (Vaswani et al., 2017; Ott et al., 2018).

Table 1: Avg. feature count ($p$) in test samples used to evaluate `Mahé`'s explanations of target models. †Superpixels are used to reduce the dimensionality of images (§4.1). ††For Transformer, $p$ and the dataset (Merity et al., 2016) are based on experimental design (§5.3).

| models | dataset | average $p$ |
|---|---|---|
| DNA-CNN | MYC-DNA | 36 |
| Sentiment-LSTM | SST | $15.9 \pm 7.0$ |
| ResNet152† | ImageNet '14 | $30.2 \pm 1.4$ |
| Transformer†† | WikiText-103 | $11.8 \pm 2.3$ |

Avg. $p$ for our context-dependent evaluations, similar to our context-free tests, are shown in Table 1.

The following hyperparameters are used in our experiments. We use $n = 1k$ local-vicinity samples in $\mathcal{D}$ for synthetic experiments and $n = 5k$ samples for experiments explaining models of real-world datasets, with 80%-10%-10% train-validation-test splits to train and evaluate `Mahé`. The distance metrics for vicinity size are: $\ell_2$ distance for synthetic experiments, cosine distance for DNA-CNN and ResNet152, and edit distance for Sentiment-LSTM and Transformer. We use on-off superpixel and word approaches to binary feature representation for explaining ResNet152 and Sentiment-LSTM respectively (Ribeiro et al., 2016; Lundberg & Lee, 2017), and the other experiments for real-world datasets use perturbation distributions that randomly perturbs features to belong to the same categories of original features, as in (Ribeiro et al., 2018).The superpixel segmenter we use is quick-shift (Vedaldi & Soatto, 2008; Ribeiro et al., 2016).

For the hyperparameters of the neural networks in `Mahé`, we use MLPs with 50-30-10 first-to-last hidden layer sizes to perform interaction detection in the NID framework (Tsang et al., 2018). These MLPs are trained with $\ell_1$ regularization $\lambda_1 = 5e{-}4$. The learning rate used is always $5e{-}3$ except for Transformer experiments, whose learning rate of $5e{-}4$ helped with interaction detection under highly unbalanced output classes. The MLP-based interaction models in the GAM (Eq. 3)

---

[2]Unfortunately, for image-related tasks, we could not generalize our definition of local interactions despite the translation invariance of deep convnets.

[3]The motif and flanking regions of DNA sequences in the training set are shuffled to simulate unalignment.

(a) interaction fit (std. MSE; lower is better)    (b) interaction detection (R-precision; higher is better)

Figure 3: Results of synthetic experiments with Mahé and different baselines explaining base models XGBoost (tree), MLP, LSTM trained on $F_{1-4}$ (Table 2) at $\sigma = 0.6$ (max$= 3.2$) are shown. (a) shows the average local fit in MSE of Mahé and baselines on the base models' representations of respective interactions. (b) shows the average R-precision of interaction rankings from each baseline. *Shap-Tree cannot detect or fit to a three-way interaction. †We assume interaction order is unknown, and ACD-MLP and ACD-LSTM require exhaustive search of all possible interactions.

always have architectures of 30-10. They are trained with $\ell_2$ regularization of $\lambda_2 = 1e{-}5$ and learning rate of $1e{-}3$. Because learning GAMs can be slow, we make a linear approximation of the univariate functions in Eq. 3, such that $g_i(x_i) = x_i$. This approximation also allows us to make direct comparisons between Mahé and linear LIME, since $x_i$ is exactly the linear part (Eq. 1). All neural networks train with early stopping, and Level $L + 1$ is decided where validation performance does not improve more than $10\%$ with a patience of 2 levels. $c$ ranges from 3 to 4 in our experiments.

## 5.2 CONTEXT-DEPENDENT EXPLANATIONS

### 5.2.1 SYNTHETIC EXPERIMENTS

In order to evaluate Mahé's context-dependent explanations, we first compare them to state-of-the-methods for local interaction interpretation. A standard way to evaluate the accuracy of interaction detection and modeling methods has been to experiment on synthetic data because ground truth interactions are generally unknown in real-world data (Hooker, 2004; Sorokina et al., 2008; Lou et al., 2013; Tsang et al.,

Table 2: Data generating functions with interactions

| | |
|---|---|
| $F_1(\mathbf{x}) =$ | $10x_1x_2 + \sum_{i=3}^{10} x_i$ |
| $F_2(\mathbf{x}) =$ | $x_1x_2 + \sum_{i=3}^{10} x_i$ |
| $F_3(\mathbf{x}) =$ | $\exp(|x_1 + x_2|) + \sum_{i=3}^{10} x_i$ |
| $F_4(\mathbf{x}) =$ | $10x_1x_2x_3 + \sum_{i=4}^{10} x_i$ |

2018). Similar to Hooker (2007), we evaluate interactions in a subset region of a synthetic function domain. We generate synthetic data using functions $F_1 - F_4$ (Table 2) with continuous features uniformly distributed between $-1$ to $1$, train complex base models (as specified in §5.1) on this data, and run different local interaction interpretation methods on 10 trials of 20 data instances at randomly sampled locations on the synthetic function domain. Between trials, base models with different random initializations are trained to evaluate the stability of each interpretation method. We evaluate how well each method fits to interactions by first assuming the true interacting variables are known, then computing the Mean Squared Error (MSE) between the predicted interaction attribution of each interpretation method and the ground truth at 1000 uniformly drawn locations within the local vicinity of a data instance, averaged over all randomly sampled data instances and trials (Figure 3a). We also evaluate the interaction detection performance of each method by comparing the average R-precision (Manning et al., 2008) of their interaction rankings across the same sampled data instances (Figure 3b). R-precision is the percentage of the top-$R$ items in a ranking that are correct out of $R$, the number of correct items. Since $F_1 - F_4$ only ever have 1 ground truth interaction, $R$ is always 1. Compared to Shap-Tree, ACD-MLP, and ACD-LSTM, the Mahé framework is the only one capable of detection *and* fitting, and it is the only model-agnostic approach.

### 5.2.2 EVALUATING ON REAL-WORLD DATA

In this section, we demonstrate our approaches to evaluating Mahé's context-dependent explanations on real-world data. We first evaluate the prediction performance of Mahé on the test set of $\mathcal{D}$ as interactions are added in Eq. 3, i.e. $K$ increases. For a given value of $\sigma$, we run Mahé 10 times on each of 40 randomly selected data instances from the test sets associated with DNA-CNN, Sentiment-LSTM, and ResNet152. For Transformer, performance is examined on a specific grammar (*cet*) translation, to be detailed in §5.3. The local vicinity samples and model initializations in Mahé are randomized in every trial. We select the $\sigma$ that gives the worst performance for Mahé at $K = L$ in each base model, out of $\sigma = 0.4\sigma'$, $0.6\sigma'$, $0.8\sigma'$, and $1.0\sigma'$, where $\sigma'$ is the average pairwise distance between data instances in respective test sets. Results are shown in Table 3 for $K$ starting from 0, which is linear LIME, and increasing to the last hierarchical level $L$.

Table 3: Average prediction performance (lower is better; 1-AUC for Transformer, MSE otherwise) with ($K > 0$) and without ($K = 0$) interactions for random data instances in the test sets of respective base models. Only results with detected interactions are shown. For each model, at least 80% of all tested data instances possessed interactions, yielding $\geq$ 320 instances for each performance statistic. Including interactions results in significant performance improvements.

|  | $K$ | DNA-CNN | Sentiment-LSTM | ResNet152 | Transformer |
|---|---|---|---|---|---|
| linear LIME | 0 | 9.8e$-$3 $\pm$ 8.8e$-$4 | 10.1e$-$2 $\pm$ 7.0e$-$3 | 0.25 $\pm$ 0.068 | 0.25 $\pm$ 0.071 |
| Mahé | 1 | 8e$-$3 $\pm$ 1.3e$-$3 | 5.6e$-$2 $\pm$ 8.6e$-$3 | 0.22 $\pm$ 0.063 | 0.06 $\pm$ 0.016 |
| Mahé | L | 6e$-$3 $\pm$ 1.2e$-$3 | 2.4e$-$2 $\pm$ 7.2e$-$3 | 0.16 $\pm$ 0.053 | 0.06 $\pm$ 0.015 |

(a) Explanation A of negative prediction

(b) Explanation B of negative prediction

Figure 4: Example of explanations that Mechanical Turk users choose from for a sentiment analysis task. (a) is linear LIME, (b) is Mahé. LIME explanations are shown as positive and negative contributions of each feature (word) to the prediction, and Mahé explanations are shown similarly with one of the contributions belonging to a single interaction or group of words.

An alternative approach to evaluating Mahé is to determine out of LIME and Mahé explanations, could human evaluators prefer Mahé explanations? We recruit a total of 60 Amazon Mechanical Turk users to participate in comparing explanations of Sentiment-LSTM predictions. While the presented LIME explanations are standard, we adjust Mahé to only show the $K = 1$ interaction and merge its attribution with subsumed features' attributions to make the difference between LIME and Mahé subtle (Figure 4). We present evaluators with explanations for randomly selected test sentences under the main condition that these sentences must have at least one detected interaction, which is the case for $> 95\%$ of sentences. In total, there are explanations for 40 sentences, each of which is examined by 5 evaluators, and a majority vote of their preference is taken. Each evaluator is only allowed to pick between explanations for a maximum of 4 sentences. Please see Appendix B for additional conditions used to select sentences for evaluators and more examples like Figure 4. The result of this experiment is that the majority of preferred explanations (65%, $p = 0.029$) is with interactions, supporting their inclusion in hierarchical explanations.

### 5.2.3 HIERARCHICAL EXPLANATIONS

Examples of context-dependent hierarchical explanations for ResNet152, Sentiment-LSTM, and Transformer are shown in Figure 6, Table 6, and Appendix E respectively after page 9. For the image explanations in Figure 6, superpixels belonging to the same entity often interact to support its prediction. One interesting exception is (Figure 6 (d)) because water is not detected as an important interaction with buffalo in the prediction of water buffalo. This could be due to various reasons. For example, water may not be a discriminatory feature because there are a mix of training images of water buffalo in ImageNet with and without water. The same is true for related classes like bison. Explanations may also appear unintuitive when a model misbehaves. Therefore, quantitative validations, such as the predictive performance of adding interactions in each hierarchical level (e.g. $R^2$ scores in Figure 6), can be critical for trusting explanations.

### 5.3 CONTEXT-FREE EXPLANATIONS

In this section, we show examples of context-free explanations of interactions found by Mahé. We first study the context-free interactions learned by Sentiment-LSTM. To have enough sentences for this evaluation, we use data from IMDB movie reviews (Maas et al., 2011) in addition to the test set of SST. Based on our results (Figure 5), we observe that the polarities of certain local interactions are almost always the same, where the words of matching interactions can be separated by any number of words in-between. To ensure that this global behavior is not a coincidence, we modify local interaction behavior in Sentiment-LSTM to check for a global change in this behavior (§4.2). As a result, when the model's local interaction attribution at a *single data instance* is negated, the attribution is almost always the opposite sign for the rest of the sentences.

Table 4: *cet* interactions before and after modifying Transformer. $N_s$ is number of samples, and %*cet* is % of $N_s$ samples + or − contributing to *cet*.

| | Before modifying | | After modifying | |
|---|---|---|---|---|
| Interaction | $N_s$ | %*cet* + | $N_s$ | %*cet* − |
| (this, event) | 38 | 1.0 | 39 | 1.0 |
| (this, article) | 33 | 1.0 | 34 | 1.0 |
| (this, incident) | 31 | 1.0 | 29 | 0.93 |
| (this, album) | 24 | 1.0 | 40 | 1.0 |
| (this, arrangement) | 22 | 1.0 | 36 | 1.0 |
| (that, afternoon) | 22 | 1.0 | 27 | 1.0 |
| (this, location) | 20 | 1.0 | 22 | 0.95 |
| (this, effect) | 19 | 1.0 | 20 | 0.95 |

Table 5: Examples of En.-Fr. translations before and after modifying Transformer. Interacting elements are bolded. BLEU change is the % change in test BLEU score from modifying the bolded interaction in Transformer.

| | Sample English-French Translations | BLEU change |
|---|---|---|
| English | **This event** took place on 10 August 2008. | |
| Fr. before | **Cet** événement a eu lieu le 10 Mars 2008. | |
| Fr. after | Cette rencontre a eu lieu le 10 Mars 2008. | (−3.7%) |
| English | **This incident** made it into the music video. | |
| Fr. before | **Cet** incident a été intégré dans le vidéo musical. | |
| Fr. after | C'est pas mal du tout ca! | (−3.4%) |
| English | The initial language of **this article** was French. | |
| Fr. before | La langue initiale de **cet** article était le Français. | |
| Fr. after | La langue originale du présent article était le Français. | (−2.8%) |

A notable insight about Sentiment-LSTM is that it appears to represent (too, bad) and (only, worse) as globally positive sentiments, and `Mahé`'s modification in large part rectifies this misbehavior (Figure 5). The modifications to Sentiment-LSTM only cause an average reduction of 1.5% test accuracy, indicating that the original learned representation stays largely intact. Results for $\sigma = 16$ are shown with the average pairwise edit distance between sentences being $\sigma' = 24.8$. Words in detected interactions are separated by 1.3 words on average.

Next, we study the possibility of identifying context-free interactions in Transformer on a known form of interaction in English-to-French translations: translations into a special French word for "this" or "that", *cet*, which only appears when the noun it modifies begins with a vowel. Some examples of *cet* interactions are (this, event), (this, article), and (this, incident), whose nouns have the same starting vowels in French. For our explanation task, the *presence* of *cet* in a translation is used as a binary prediction variable for local interaction extraction. To minimize the sources of *cet*, we limit original sentence

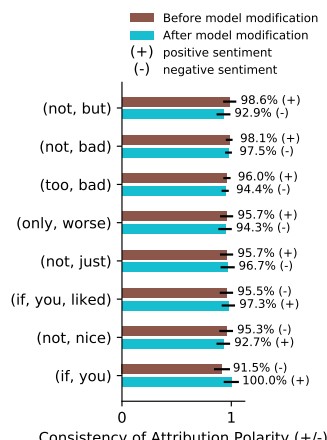

Figure 5: Interaction polarity consistency in Sentiment-LSTM before and after model modification. (mean and std. errors shown)

lengths to 15 words, and we perform translations on WikiText-103 (Merity et al., 2016) to evaluate on enough sentences. The results of context-free experiments on *cet* interactions of adjacent English words are shown in Table 4. The interactions always have positive polarities towards *cet*, and after modifying Transformer at a single data instance for a given interaction, its polarity almost always become negative, just like the context-free interactions in Sentiment-LSTM. Examples of new translations from the modified Transformer are shown in the "after" rows in Table 5, where *cet* now disappears from the translations. The test BLEU score of Transformer only decreases by an average percent difference of −2.7% from modification, which is done through differentiating the max value of *cet* output neurons over all translated words. Results for $\sigma = 6$, $\sigma' = 10.5$ are shown.

Experiments on DNA-CNN and ResNet152 show similar results at fixed interaction positions (§4.2). For DNA-CNN, out of the 94 times a 6-way interaction of the CACGTG motif (Sharon et al., 2008) was detected in the test set, every time yielded a positive attribution polarity towards DNA-protein affinity, and the same was true after modifying the model in the opposite polarity (cosine distance $\sigma = 0.35$, $\sigma' = 0.408$). For ResNet152, context-free interactions are also found (cosine distance $\sigma = 0.4$, $\sigma' = 0.663$). However, because superpixels are used, the interactions found may contain artifacts caused by superpixel segmenters, yielding less intuitive interactions (see Appendix A).

## 5.4 LIMITATIONS

Although `Mahé` obtains accurate local interactions on synthetic data using NID, there is no guarantee that NID finds correct interactions. `Mahé` faces common issues of model-agnostic perturbation methods in interpreting high-dimensional feature spaces, choice of perturbation distribution, and speed (Ribeiro et al., 2016; 2018). Finally, an exhaustive search is used for context-free explanations.

# 6 CONCLUSION

In this work, we proposed Mahé, a model-agnostic framework of providing context-dependent and context-free explanations of local interactions. Mahé has demonstrated the capability of outperforming existing approaches to local interaction interpretation and has shown that local interactions can be context-free. In future work, we wish to make the process of finding context-free interactions more efficient, and study to what extent model behavior can be changed by editing its interactions or univariate effects. Finally, we would like to study the interpretations provided by Mahé more closely to find new insights into structured data.

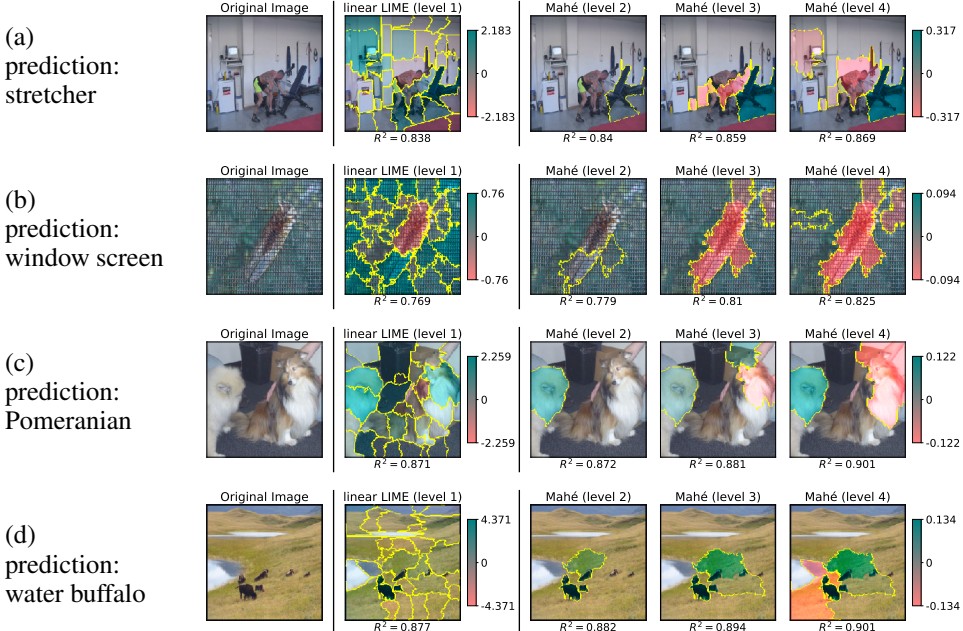

Figure 6: Examples of context-dependent explanations in hierarchical format for ResNet152, where images come from the ImageNet test set. Interaction attributions of $\{g'_i(\cdot)\}_{i=1}^{K}$ are show at each $K-1$ level, $K \geq 1$ (§4.1). Colors in superpixels represent attribution scores and their polarity. Cyan regions positively contribute to the predicton, and red regions negatively contribute. Boundaries between overlapping interactions are merged when their attribution polarities match.

Table 6: Examples of context-dependent hierarchical explanations on Sentiment-LSTM. The interaction attribution of $g'_K(\cdot)$ is shown at each $K-1$ level, $K \geq 1$ (§4.1) in color. Green means positively contributing to sentiment, and red the opposite. Visualized attributions of linear LIME and Mahé are normalized to the max attribution magnitudes (max magn.) shown. Top-5 attributions by magnitude are shown for LIME.

| Method | Level | Fit ($R^2$) | Hierarchical Explanation | Max magn. |
|---|---|---|---|---|
| linear LIME | 1 | 0.621 | the film is really not so much bad as bland | 0.744 |
| Mahé | 2 | 0.751 | not, bad | |
| Mahé | 3 | 0.916 | not, bad, bland | |
| Mahé | 4 | 0.926 | film, not, bad, bland | 0.119 |
| linear LIME | 1 | 0.519 | a very average science fiction film | 0.708 |
| Mahé | 2 | 0.598 | science, fiction | |
| Mahé | 3 | 0.819 | a, average | |
| Mahé | 4 | 0.923 | a, very, average | 0.213 |
| linear LIME | 1 | 0.612 | a charming romantic comedy that is by far the lightest dogme film and among the most enjoyable | 0.612 |
| Mahé | 2 | 0.856 | charming, enjoyable | |
| Mahé | 3 | 0.923 | charming, lightest, enjoyable | 0.072 |

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

## SUPPLEMENTARY MATERIALS

## A    CONTEXT-FREE EXPLANATIONS IN RESNET152

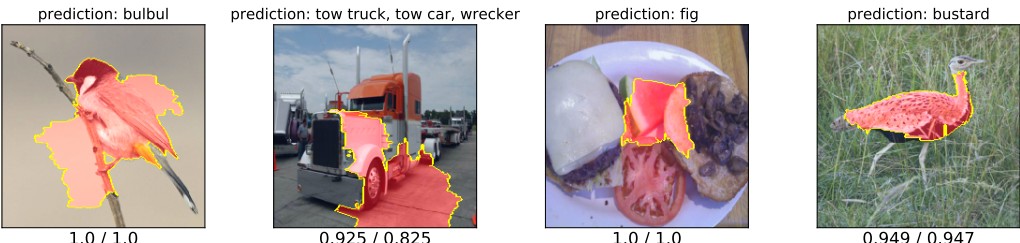

Figure 7: Context-free explanations of interactions in ResNet152 for each image, evaluated over 40 tests before and after modification. Each test superimposes the interaction of interest onto a randomly selected background image from the test set of ImageNet. Corresponding predictions are shown above each image, and the percentage of consistent interaction polarity before and after modification are shown below in that order. The red color indicates a negative interaction attribution polarity before modification. After modification, the polarities become positive.

## B    FURTHER DETAILS OF MECHANICAL TURK EXPERIMENT

Besides requiring detected interactions, several other conditions were used to choose sentences for Mechanical Turk evaluators. We ensure that there is a significant attribution difference between LIME and `Mahé` by only choosing among sentences that have a polarity difference between `Mahé`'s interaction and LIME's corresponding linear attributions. To reduce ambiguities of uninterpretable explanations arising from a misbehaving model - an issue also faced by Sundararajan et al. (2017) in interpretation evaluation - we only show explanations of sentences that the model classified correctly. We also attempt to limit the effort that evaluators need to analyze explanations by only showing sentences with 5-12 words with uniform representation of each sentence length.

An example of the interface that evaluators select from is shown in Figure 8. Figure 9 shows randomly selected examples that evaluators analyze. The visualization tool for presenting additive attribution explanations is graciously provided by the official code repository of LIME [4].

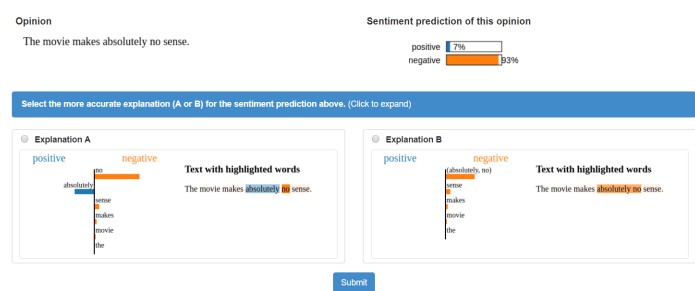

Figure 8: Example of Mechanical Turk interface used by workers to select between explanations provided by linear LIME and `Mahé`.

---

[4]https://github.com/marcotcr/lime

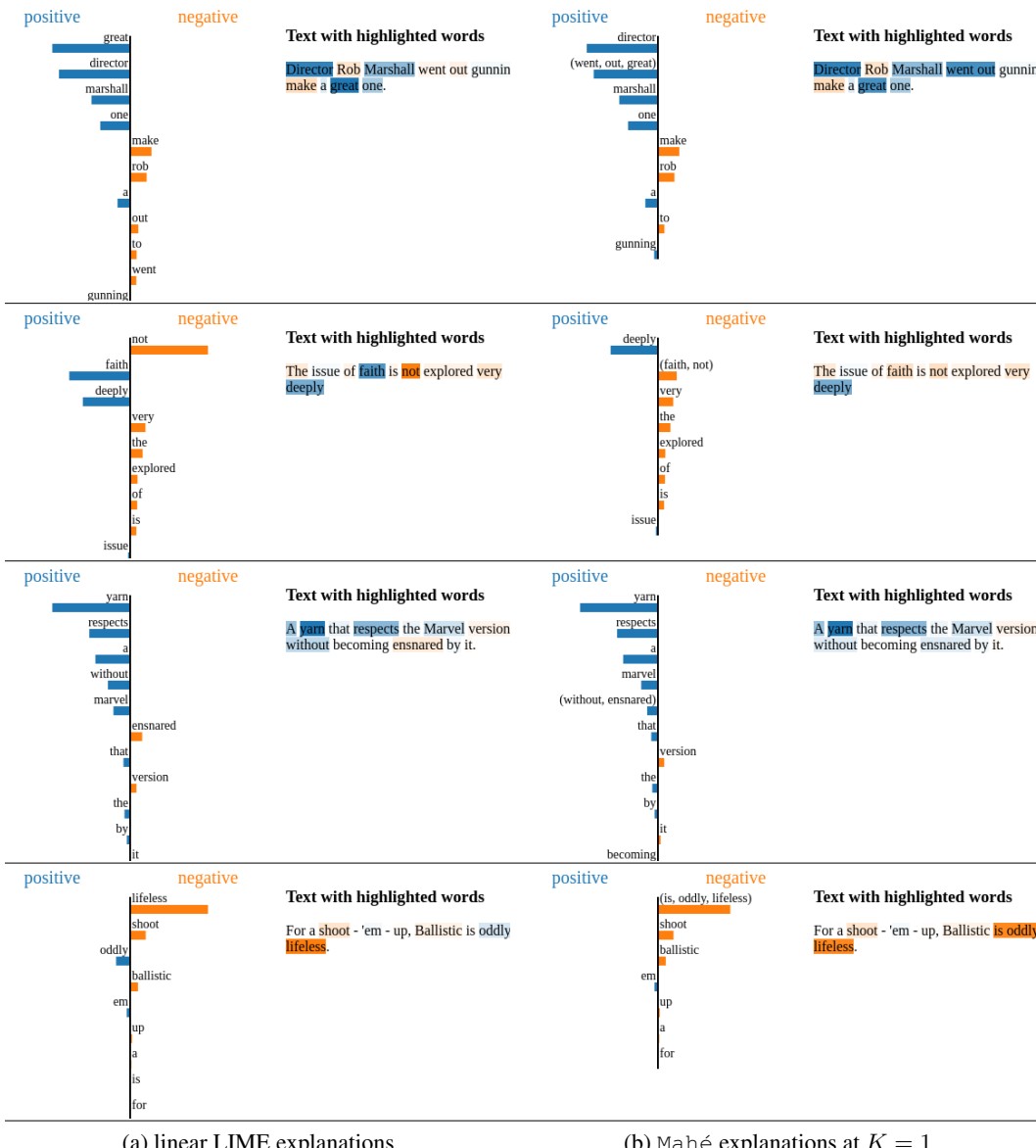

(a) linear LIME explanations  (b) Mahé explanations at $K = 1$

Figure 9: Randomly selected comparisons between (a) linear LIME and (b) Mahé explanations of Sentiment-LSTM used in Mechanical Turk experiments (§5.2.2).

## C   RUNTIME

Figures 10 and 11 show runtimes of context-dependent and -free explanations using Mahé. All experiments were conducted on Intel Xeon 2.4-2.6 GHz CPUs and Nvidia 1080 Ti GPUs. Experiments with MLPs were run on CPUs and inference/retraining of DNA-CNN, Sentiment-LSTM, ResNet152, and Transformer were run on GPUs.

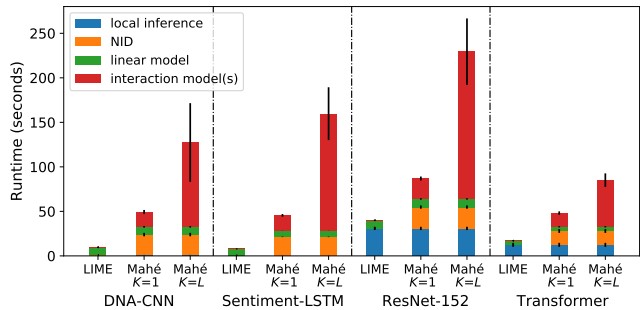

Figure 10: Average runtime of linear LIME versus Mahé on context-dependent explanations. Runtimes for experiments in Table 3 are shown. "local inference" is the runtime for sampling in the local vicinity of a data instance and running inference though a black-box model for every sampled point. "NID" is the runtime for running NID interaction detection. "linear model" is the runtime for training a linear model (Eq. 1) to get linear attributions with LIME. "interaction model(s)" is the runtime for sequentially training interaction models (Eq. 3) to get interaction attributions with Mahé.

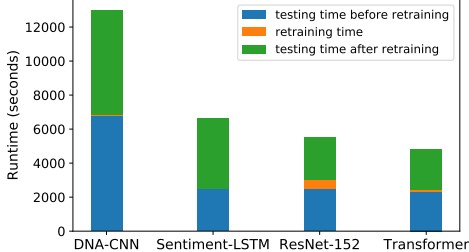

Figure 11: Runtime of Mahé for determining whether an interaction is context-free for a randomly selected interaction and 40 different contexts, run sequentially. Runtimes for checking interaction consistency before and after model retraining (fine-tuning) are shown, resulting in tests on 80 contexts total. DNA-CNN takes longer here because we needed to relax the cutoff criteria of identifying the last hierarchical level to find the CACGTG interaction. For context-free experiments, a cutoff patience (§5.1) for Sentiment-LSTM, ResNet152, and Transformer was not needed in our experiments and is excluded in this runtime analysis. The patience for DNA-CNN was 2.

# D    COMPARISONS TO BASELINES FOR CONTEXT-FREE EXPLANATIONS

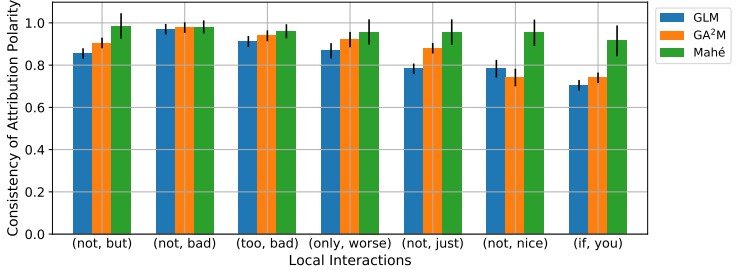

Figure 12: Comparisons of Mahé to baselines for identifying consistent interaction polarity before modifying models. Results for explaining Sentiment-LSTM are shown on the same interactions identified in Figure 5. The baselines are GLM and GA$^2$M. GLM is a lasso-regularized generalized linear model with all pairs of multiplicative interaction terms (Bien et al., 2013), and GA$^2$M is a tree-based generalized additive model with pairwise non-additive interactions (Lou et al., 2013).

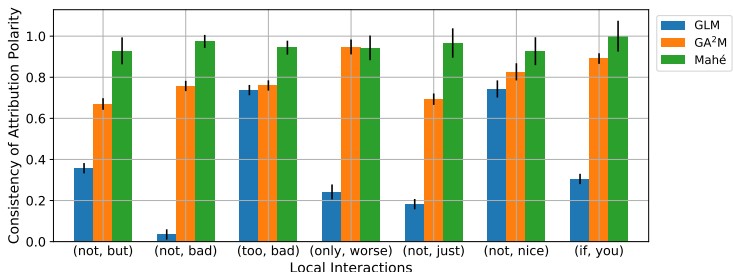

Figure 13: Comparisons of Mahé to baselines for identifying consistent negated interaction polarity after using the same baselines to locally modify Sentiment-LSTM on the interactions from Figure 12. Mahé shows more significant improvements over baselines than before negating interactions.

## E  HIERARCHICAL EXPLANATIONS OF *cet* INTERACTIONS IN TRANSFORMER

Table 7: Examples of context-dependent hierarchical explanations on Transformer. The interaction attribution of $g'_K(\cdot)$ is shown at each $K - 1$ level, $K \geq 1$ (§4.1) in color. Green contributes towards *cet* translations, and red contributes the opposite. Visualized attributions of linear LIME and Mahé are normalized to the max attribution magnitudes (max magn.) shown. Top-5 attributions by magnitude are shown for LIME.

| Method | Level | Fit ($R^2$) | Hierarchical Explanation | Max magn. |
|---|---|---|---|---|
| linear LIME | 1 | 0.782 | this article was last updated on substance in august 2012 | 0.657 |
| Mahé | 2 | 0.948 | this, article | 3.707 |
| linear LIME | 1 | 0.696 | this effect takes part in making lead slightly less reactive chemically | 0.643 |
| Mahé | 2 | 0.96 | this, effect | 2.459 |
| linear LIME | 1 | 0.734 | the population size of this bird has not yet been quantified or estimated | 0.605 |
| Mahé | 2 | 0.926 | this, bird | 1.211 |

## F  EXPERIMENTS WITH LARGE NUMBER OF FEATURES

We performed experiments on the accuracy and runtime of the MLP used for interaction detection (via NID) on datasets with large number of features. We generate synthetic data of $n$ samples and $p$ features $\{(\mathbf{X}^{(i)}, y^{(i)})\}$ with randomly generated pairwise interactions of using the following equation (Purushotham et al., 2014):

$$y^{(i)} = \boldsymbol{\beta}^\top \mathbf{X}^{(i)} + \mathbf{X}^{(i)\top} \mathbf{W} \mathbf{X}^{(i)},$$

where $\mathbf{X}^{(i)} \in \mathbb{R}^p$ is the $i^{th}$ instance of the design matrix $\mathbf{X} \in \mathbb{R}^{p \times n}$, $y^{(i)} \in \mathbb{R}$ is the $i^{th}$ instance of the response variable $\mathbf{y} \in \mathbb{R}^{n \times 1}$, $\mathbf{W} \in \mathbb{R}^{p \times p}$ contains the weights of pairwise interactions, $\boldsymbol{\beta} \in \mathbb{R}^p$ contains the weights of main effects, and $i = 1, \dots, n$. $\mathbf{W}$ was generated as a sum of $K$ rank one matrices, $\mathbf{W} = \sum_{k=1}^{K} \boldsymbol{a}_k \boldsymbol{a}_k^\top$. $\mathbf{X}$ is normally distributed with mean 0 and variance 1. Both $\boldsymbol{a}_k$ and $\boldsymbol{\beta}$ are sparse vectors of $2 - 3\%$ nonzero density and are normally distributed with mean 0 and variance 1. $K$ was set to be 5.

We found that in low $p$ settings, i.e. $p = 100$, $n$ only needed to be at least $10p$ to recover 5-15 pairwise interactions at AUC$> 0.9$. Increasing $p$ to 1000 still required $n > 10p$, but performance stability significantly improved between $10p$ and $100p$ for detecting 900-2000 interactions. When $p = 10k$, we could not detect interactions at $n = 10p$ and did not study further due to large training time. In general, increasing $n$ by an order of magnitude at fixed $p$ required 4-9x more runtime. As a rough estimate, increasing $p$ by an order of magnitude at fixed $n$ required 2-3x more runtime. There is high variance in the runtime associated with increasing $p$ because of the early stopping used.

Based on our experiments, we recommend limiting $p$ to be under 100, so that model training can complete in under 40 seconds. Once interaction detection via NID is done, the extracted interaction

sets tend to be much smaller than $p$, and $\phi_K$ (Eq. 3) for each interaction is likely to train faster than the original MLP with $p$ inputs. We note that identifying interactions in high dimensional input spaces like images and image models is an interesting and challenging research problem and is left for future work.

## G    MORE EXAMPLES OF INTERACTIONS WITH CONSISTENT POLARITIES IN SENTIMENT-LSTM

Table 8: Shown below are more examples of interactions with consistent polarities found by Mahé in Sentiment-LSTM. Num samples is the number of sentences from which the same interaction is found. Percent polarity is the percentage of interactions that have the specified attribution polarity. Avg. separation is the average separation of words in detected interactions.

| Interaction | num samples | percent polarity | | avg. separation between words |
|---|---|---|---|---|
| (not, good) | 213 | 96.7% | negative | 1.4 |
| (falls, flat) | 169 | 97.6% | negative | 0.17 |
| (not, funny) | 155 | 97.4% | negative | 0.65 |
| (not, miss) | 133 | 97.7% | positive | 0.66 |
| (still, love) | 103 | 98.1% | positive | 0.19 |
| (bad, worst) | 44 | 95.5% | positive | 5.5 |
| (never, off) | 36 | 100% | positive | 1.4 |

