# OpenReview forum: "Can I trust you more? Model-Agnostic Hierarchical Explanations"
_ICLR.cc/2019/Conference_

### Official Review · AnonReviewer1 · 2018-10-23
**Clear methodological contribution but not written clearly enough**

**Rating:** 5
**Confidence:** 4

**Review:**

Summary
=======
The authors extended the linear local attribution method LIME for interpreting black box models by non-linear functions to more accurately approximate black box models locally and identifying interactions between model input variables using the previously published neural interaction detection (NID) framework. They further propose a method to discern between context-dependent and context-free interactions. I found the paper hard to understand without being familiar with previously published literature on detecting interactions and could not understand their approach to detect context-free interactions as well as some aspects of their evaluation. I have also concerns about the practically of their method due to the high runtime and the notion of locality in the light of high-dimensional inputs. In the following, I will briefly summarizing my major criticism and give further details below.

Summary of major criticism
=====================
1) The paper is hard to understand without being familiar with previously published literature in the field.
The authors do not describe how they define the interaction sets X_I in equation (3).

2) I could not understand their approach for detecting context-free interactions (section 4.2).
3) It is unclear how discrete variables are locally modified. Can the approach be used for a combination of differently distributed variables, e.g. categorical and continuous variables?
4) Evaluation metrics such as MSE and R-precision are not described and I could not understand other important aspects of their evaluation, e.g. superpixels (figure 6), why and how they modified network architectures for evaluating context-free interactions (section 5.3), or how they asked Amazon Medical Turk users.
5) The authors did not compare the runtime of Mathe with LIME, which is presumably high due to the need of fitting multiple non-linear models (equation 3) and sampling the local neighborhood.
6) The authors  did not compare the total number of model parameters of Mathe with LIME, which might also account for the higher accuracy (lower MSE).
7) The authors did not evaluate the accuracy and runtime of Mathe on high-dimensional inputs, e.g. large images.


Details
=====

Abstract
---------------------------
1. The abstract is hard to understand since ‘context-dependent’ and ‘context-free’ are undefined. The authors should also not use ‘dependencies’ as synonym for ‘interactions’.

Introduction
---------------------------
2. The difference between interactions and context-free and context dependent interactions is unclear. Is a variable without interactions context-free, e.g. Buffalo does not interact with water, and a variable with interactions context-dependent? Also, ‘classes of data’ is unclear, which can be misinterpreted as class labels for classification problems. The authors should also clarify what they mean by ‘performance and generality’ in the last section of the introduction.

Section 3.1
---------------
3. The description of the model function f(x) and attribution scores phi(x) is unclear. Since f(x) is undefined when it is first mentioned after equation (1), I suggest to first define f(x) and afterwards define phi(x). The purpose of the attribution score function phi(x) is also unclear without prior knowledge. The authors should more clearly describe that f(x) is the target function (model) of interest, e.g. a classifier, and phi(x) locally approximates f(x) and is interpretable in contrast to f(x).

Section 4.1
---------------
4. The authors should justify why they are sampling x ~ N(x, sigma * I), which assumes that data instances are iid Normal. This is not the case, e.g., if x is categorical or contains a combination of categorical and continuous variables, and variables are correlated. How is sigma chosen? How many samples are used depending on the dimension of x?

5. The authors should briefly describe the basic idea of NID.

6. How are points sampled from the epsilon neighborhood of x? What is a link function?

7. The subsection ‘Hierarchical Interaction Attribution’ is hard to understand without being familiar Tsang et al. The authors should give an example of a hierarchical explanation with different layers.

Section 4.2
---------------
8. How is the ‘local vicinity’ defined? Which distance metric is used? This is in particular problematic if x is high-dimensional due the the curse of dimensionality. How are continuous and categorical variables locally modified? Did the authors meant to use a lowercase ‘k’ in equation (4), i.e. ‘phi_k =’ instead of ‘phi_K’? I find this section hard to understand without being familiar with the cited literature.

Section 5.1
---------------
9. The number of local vicinity samples is unclear. Did the authors use 1k local vicinity samples for synthetic experiments (Table 1) and 5k samples for real-word synthetic experiments?

10. What is the dimensionality (number of words, characters, or pixels) of real world datasets? This is important since it influences the number samples that are required to approximate the vicinity of a particular data point. It is in particular interesting to know how the model accuracy and runtime depends on the dimensionality and the number of local vicinity samples.

11. The authors should define the evaluation metrics (MSE, R-precision) in addition to citing them.

12. How did the authors choose the interaction sets X_I in equation (3) and (4)? How many MLPs (functions g(.)) did the authors fit to learn phi(x)? Is the number of MLPs the same for LIME and Mathe? Otherwise the performance gain of Mathe over LIME can also be attributed the increased number of models and model parameters (ensemble size).

13. What is the average training time of Mathe and baseline models on the different datasets?

Section 5.2.2
-----------------
14. How did the authors choose sigma (0.4, 6, 0.4) for the different datasets?

15. How did Amazon medical turk users evaluate Mathe vs. LIME interactions? Were they given for each sentence the best Mathe and best LIME interaction and asked to decide which one is better? Figure 4 should be more clearly described in the caption. The sentence ‘The result of this experiment is that the majority of preferred explanation …’ is unclear and unjustified by only showing one example in Figure 4.

Section 5.2.3
-----------------
16. The authors should discuss figure 6. The results indicate that neither LIME nor Mathe is able to clearly identify the object of interest, e.g. the water buffalo, and interactions, e.g. between the buffalo and water.

Section 5.3
---------------
17. The sentence ‘... the presence of a French word for “this” or “that”, cet, which …’ is unclear. I suggest to give an example to illustrate which interactions are supposed to be detected. The modification of the Transformer model and the reason why this is necessary is unclear. Overall, I find this evaluation unclear and insufficient since it only applies to a particular interaction.

---

> ### Author Response · Authors · 2018-11-08
> **Early Author Response**
>
> Dear reviewer,
>
> We are in the process of performing experiments and improving presentation based on your suggestions, but we would like to request clarification in advance about your difficulty understanding our paper. We have added a brief description of the previous literature of NID in Section 4.1 (in blue) and was wondering if this is the type of clarification you are looking for. In addition, we are wondering what part of our Context-Free approach in Section 4.2 is hard to understand? In this section, the cited literature appears after we discuss the bulk of our approach.
>
> MSE and R-Precision are standard metrics in Machine Learning. Superpixels for image models are also standard in model-agnostic explanation methods like LIME, Shap, and Anchors. Section 5.2.2 referred to details about the Mechanical Turk experiment in Supplementary B, which was included in our original submission.

---

> > ### Comment · AnonReviewer1 · 2018-11-12
> > **Response**
> >
> > Thanks for outlining the basic idea of NLP. What does ‘efficiently’ mean, i.e. what is the time complexity of NLP compared with O(2^p)?
> >
> > Please see ‘Section 4.2’ in my main review for what I find unclear about this section.
> >
> > I agree that MSE = 1/n (a - b)^2 is widely used in machine learning but it is unclear what a and b are. While ‘Precision’ is well known, ‘R precision’ is ranking-specific and should be defined. Superpixels might be established in the context of model-agnostic explanation, but should be also understandable for readers who are unfamiliar with previous literature in this domain. Please briefly explain and cite superpixels and how they were computed.

---

> ### Author Response · Authors · 2018-11-24
> **Author Response (2/2)**
>
>
> Section 5.1 (continued):
>
> The average runtime between Mahe and LIME for the different datasets has been added in Figure 10 in Appendix C, which includes runtimes for how long sampling and fitting nonlinear models take.
>
> Based on your suggestion, we performed experiments on the accuracy and runtime of the MLP used for interaction detection on large p data. We generate synthetic data with randomly generated pairwise  interactions using the equation y^(i) = X^(i)WX^(i) + beta*X^(i) (Purushotham et al. 2014). X^(i) are rows of dimension p in a nxp design matrix, and W is a pxp matrix that indicates which variables will interact. We set a 2-3% nonzero density for the generation of W (Purushotham et al. 2014). We found that in low p settings, p=100, n only needed to be at least 10p to recover 5-15 pairwise interactions at AUC>0.9. Increasing p to 1000 still required n>10p, but performance stability significantly improved between 10p and 100p for detecting 900-2000 interactions. When p = 10k, we could not detect interactions at n=10p and did not study further due to large training time. In general, increasing n by an order of magnitude at fixed p required 4-9x more runtime. As a rough estimate, increasing p by an order of magnitude at fixed n required 2-3x more runtime. There is high variance in the runtime associated with increasing p because of the early stopping used.
>
> We should emphasize that p should be held small whenever possible, such as using superpixels instead of individual pixels in images. The identification of interactions in high dimensional input spaces like images and image models is a challenging research problem and is left for future work.
>
> ------------
> Section 5.2.2: We have redone the prediction experiments (Table 3), now with a parameter sweep on sigma. Let s’ be the average pairwise distance between data instances in the test sets corresponding to respective models, provided distance metrics defined in section 5.1. Sigma is chosen to be 0.4s’, 0.6s’, 0.8s’, and 1.0s’ to represent the locality of a data instance relative to others.
>
> We have made some updates to the mechanical turk section to be clearer about what and how mechanical turk users are asked, as well as the scope of the experiment. More examples like Figure 4, now randomly selected, have been added in Appendix B.
>
> ------------
> Section 5.2.3: We have added a discussion about the water buffalo interaction in the Figure 6 caption, which is indeed interesting. We searched through examples of water buffalo in ImageNet from the link [1] and found that many images of water buffalo did not have water, and when looking at images in adjacent classes like bison, there were also many images with and without water. This leads us to question the extent that water is a discriminatory feature. The lack of a water interaction may also be caused by a misbehaving ResNet, as its top-1 error is large (22%), or a misbehaving Mahe. Unfortunately, we cannot determine whether this image was misclassified because it belongs to the ImageNet test set without test labels.
>
> ------------
> Section 5.3: We agree that introducing this section with the cet interaction can be confusing. Therefore we have rearranged this section to start with our results on sentiment analysis. The purpose of our cet experiments was to determine whether it is possible to identify context-free interactions in Transformer. We have clarified this goal in the paragraph discussing the Transformer experiment. We have also included examples of expected interactions and added the reason why we modify models to make our motivations clearer. We are actively searching for other interesting context-free interactions in Transformer and are happy to report any additional results. Currently, we are still trying to better understand French.
>
> ------------
> Regarding your follow-up suggestions:
>
> The time complexity of NID has been included in Section 4.1.
> The inputs to MSE and R-precision are now defined in Section 5.2.1.
> Superpixels and the rationale behind them are now discussed in Section 4.1 at the end of the “Local Interaction Detection” subsection. The choice of superpixel segmenter  has been added to Section 5.1.
>
>
> References:
>
> Yin Lou, Rich Caruana, Johannes Gehrke, and Giles Hooker. Accurate intelligible models with pairwise interactions. In KDD, 2013.
>
> Sanjay Purushotham, Martin Renqiang Min, C-C Jay Kuo, and Rachel Ostroff. Factorized sparse learning models with interpretable high order feature interactions. In KDD, 2014.
>
> Marco Tulio Ribeiro, Sameer Singh, and Carlos Guestrin. Why should i trust you?: Explaining the predictions of any classifier. In KDD 2016.
>
> Daria Sorokina, Rich Caruana, Mirek Riedewald, and Daniel Fink. Detecting statistical interactions with additive groves of trees. In ICML, 2008.
>
> Michael Tsang, Dehua Cheng, Yan Liu. Detecting Statistical Interactions from Neural Network Weights. In ICLR, 2018.
>
> [1] http://image-net.org/challenges/LSVRC/2014/browse-synsets

---

> > ### Comment · AnonReviewer1 · 2018-11-27
> > **Manuscript improved but is still hard to understand**
> >
> > I appreciate that you addressed all my comments tried to clarify the manuscript. Unfortunately, many parts of the manuscript remain to be hard to understand, partly due to the writing style. I therefore decided to not change the rating of the manuscript. Details below.
> >
> > Introduction
> > —————-
> > 2. Do you mean by ‘irrespective of data instance.’ ‘irrespective of data instances.’ or ‘irrespective of a particular data instance.’?
> >
> > Section 4.1
> > —————
> > I suggest to replace ‘\sigma , 1 standard deviation’ by ‘\sigma , i.e. one standard deviation’.
> >
> > Section 4.2
> > —————
> > The sentence ‘The advantage of checking the response…’ is hard to read. I suggest to shorten it by replacing verbose words (e.g. ‘whether’ by ‘if, ‘oftentimes’ by ‘often’), and splitting it into multiple sentences. Do you mean by ‘1)’ ‘definition 1)’? Which distance metrics did you use for non-textual data? How sensitive are results with respect to the distance metric? I find this paragraph is still hard to understand.
> >
> > Section 5.1
> > —————
> > What is p? The section is hard to understand since p and k are not defined in this section and the reader has to look up their definition. Please spell out indices throughout the manuscript, e.g. ‘n samples’ or ‘k features’.
> >
> > What does ‘*Superpixel’ mean?
> >
> > Thanks for describing interaction sets at the end of section 3. However, this sentence is hard to understand due to its length and technical jargon.
> >
> > I appreciate your additional analysis for different numbers of features p. Can you include and discuss the analysis in the manuscript? This is in my eyes important for being able to assess if the method is applicable for  identifying interactions in images on a pixel level (instead of superpixels).
> >
> > Section 5.2
> > —————
> > The sentence ‘We evaluate the performance of fitting ...‘ is hard to understand. Please shorten and clarify.
> >
> > Section 5.2.2
> > —————-
> > The sentence ‘We average prediction error…’ contains grammatical errors.
> >
> > Did you average sigma also for other methods? This is similar to ensembling which can increase the prediction performance.
> >
> > Thanks describing the Medical Turk experiment in an additional paragraph. Unfortunately, this paragraph also contain grammatical errors which makes it hard to understand.
> >
> > Section 5.2.3
> > —————---
> > Please discuss the results in the main text instead of the figure caption. The interpretation of subfigure d) is unfortunately not quantitative enough and it remains unclear if LIME is superior to other methods for detecting interactions in images.
> >
> > Section 5.3
> > —————-
> > The reason why you modified the transformer model and how this impacts results remains unclear. Overall, I find the evaluation is still hard to understand.

---

> > > ### Author Response · Authors · 2018-11-27
> > > **Author Response**
> > >
> > > Dear reviewer,
> > >
> > > Thank you for your additional suggestions. We have made revisions to the manuscript accordingly.
> > >
> > > ------------
> > > Introduction: we mean "irrespective of data instances"
> > >
> > > ------------
> > > Section 4.1: thanks for the suggestion
> > >
> > > ------------
> > > Section 4.2: we have split the sentence in two. "1)" has been replaced with description for clarity. Verbose words have been removed
> > >
> > > at the end of this section, we have referred to the distance metrics defined in Section 5.1 and clarified that our results are not very sensitive to exact choice of valid distance metric.
> > >
> > > ------------
> > > Section 5.1: indices have been clarified. Asterisk has been replaced with dagger for clarity. The technical jargon "iff" has been spelled out, and the sentence length has been reduced.
> > >
> > > large-dimension experiments and discussion have been added in Appendix F.
> > >
> > > ------------
> > > Section 5.2: the sentence has been shortened and clarified
> > >
> > > ------------
> > > Section 5.2.2:
> > >
> > > We have fixed the grammatical error, and we have removed the average across sigmas. For each base model, we select the sigma for which Mahe’s performance is the worst at K=L and use the same sigma for the LIME method.
> > >
> > > We have revised the mechanical turk paragraph for grammar.
> > >
> > > ------------
> > > Section 5.2.3: The discussion has been moved to the paper body. The need for quantification is now highlighted in this section. To the best of our knowledge, our approach Mahe is the only one that quantifies the predictive performance of local interaction explanations at each hierarchical step.
> > >
> > > ------------
> > > Section 5.3 : Our explanation of modifying models is introduced in the first paragraph of this section with Sentiment-LSTM. The Transformer paragraph has been revised for clarity. The context-free interactions for Transformer are now related to those of Sentiment-LSTM for continuity.

---

> > > > ### Comment · AnonReviewer1 · 2018-12-02
> > > > **Final comment**
> > > >
> > > > Thanks for your final revision: the manuscript improved. I increased my rating to 5.

---

> ### Author Response · Authors · 2018-11-24
> **Author Response (1/2)**
>
> Dear reviewer,
>
> Thank you for your thorough suggestions and follow-up response on improving our paper. We have updated many parts of the paper to improve its presentation and address your concerns.
>
> Below, we summarize our updates based on your suggestions. These updates are shown in blue in the latest paper revision as of Nov 23rd.
>
>
> ------------
> Abstract: We have clarified what is meant by “context-dependent” and “-free” and replaced “dependency” with “interaction”.
>
> ------------
> Introduction: We have clarified context-dependent and -free “interactions” as local and global interactions, and what we mean by global behavior. “Performance and generality” has been clarified to interaction fitting and detection performance and model-agnostic generality.
>
> ------------
> Section 3.1: Thanks, we have incorporated your feedback.
>
> ------------
> Section 4.1: We made a significant revision to this section. Our explanation of sampling procedure now covers continuous, categorical, and a mix of continuous and categorical data. Specifics mostly follow the procedures of previous work (Riberio et al. 2016) with some modification to the sampling kernel and local vicinity boundary. Regarding correlations, their absence can actually be advantageous for interaction detection. Suppose that two variables are correlated and one of them should naturally interact with a third variable w.r.t. an outcome variable. Because interaction signals can spread between the correlated variables, the interaction effect with the third variable is weakened, making it difficult to detect this interaction (Sorokina et al. 2008). When there are no correlations, interaction detection can focus more on identifying the true interactions in the data-generating function, i.e. the target model f().
>
> We recommend that sigma, which is used as epsilon for epsilon-neighborhood as you rightly mentioned, should be chosen based on factors such as the stability and interaction orders of explanations. If a local interaction explanation is extremely high-order and uninformative because the local vicinity covers too much of f’s complex representation, the vicinity size should be reduced. The number samples n should be larger than the feature dimension, p, to prevent the curse of dimensionality, as you mentioned. It is better to reduce p, e.g. with feature selection (or superpixel segmentation) to avoid overfitting as much as possible. We have clarified this consideration in the paper.
>
> We have provided more context and guidance on hierarchical interaction visualizations with appropriate reference to an example.
>
> ------------
> Section 4.2: Thanks for reiterating the importance of clear definitions of local vicinity and sampling procedures. We have added a reference to our new definitions in Section 4.1 and distance metric considerations at the end of this section. We have also made some clarifications about why we modify the target function f. This is for limiting the scenario that globally consistent interaction behavior is a coincidence, which is more likely to be an issue with limited samples to test on. Therefore, we modify f in part to speed up the search for context-free interactions.
>
> phi_k is now used, thank you
>
> ------------
> Section 5.1: We have made clarifications here about our number of vicinity samples, dataset dimensionality, and evaluation metrics. For MSE, we have added clarification that 1000 uniformly drawn samples within the vicinity of a data instance are used to compute interaction outputs using Mahe and baselines. The MSE is evaluated across these samples w.r.t ground truth synthetic interactions.
>
> Thanks for asking for how interaction sets are defined. We have added this definition at the end of Section 3. The definition of non-additive interaction directly tells us that it adds to the representation of a function compared to one without the interaction. This added representation is responsible for improving prediction performance, as previous works (including NID) found to have consistently approximated the performance of state-of-the-art complex models (Lou et al., 2013, Tsang et al., 2018). The MLP architecture sizes we used were already noted in Section 5.1 and are smaller (with less parameters) than those used by Tsang et al. (2018).

---

### Official Review · AnonReviewer2 · 2018-11-03
**Promising idea, but results could be better.**

**Rating:** 6
**Confidence:** 4

**Review:**

Summary of the paper:
The authors propose a framework called Mahe that provides context-dependent and context-free explanations for a given (neural network) model’s prediction. Context-dependent explanations are found by applying NID (Tsang et al, 2018) on a set of data points sampled from a neighborhood around the given input point. Further, the generalized additive model representing the function approximation around the given input is incrementally built by selectively computing higher-order-interaction terms using NID again. Each such added term results in an explanation at a level in the hierarchy. Context-free explanations are generated in two ways: 1) when a local explanation shows same polarity among all valid data points, and 2) by negating the local explanations’ polarity at a data point, fine-tuning the model on the resulting modified function approximation, and regenerating the local explanations for other data points; if the polarity is reversed for all other data points, then the local explanation is also a global explanation

Strengths:
- Broadens the application of NID to provide hierarchical explanations and context-free explanations
- Experiments on context-free explanations show promising results, for instance, on the Sentiment-LSTM model and in Supplementary A. Would be great to see more results on this front.

Questions for authors:
- The experimental results only show that using higher order interactions results in a better function approximation (explanation), but explanations for level > 2 do not seem to be that good (Table 5). For the image example, they look slightly better.
- The contribution seems incremental, given that Tsang et al (2018) already explored explanations based on interactions.

Conclusion
Considering that the NID idea has been broadened to context-free explanations, the paper shows promise, but it is a weak accept because the other contributions do not seem fully worked out.

---

> ### Author Response · Authors · 2018-11-25
> **Author Response**
>
> Dear reviewer,
>
> Thank you for understanding our work and acknowledging its strengths. Below are our responses to your questions.
>
> >> “but explanations for level > 2 do not seem to be that good (Table 5)”
>
> We have updated Table 5 (now Table 6 in the latest revision) with examples where the R^2 performance gains of levels>2 over level=2 are more significant. Previously, most examples only showed R^2 improvements of at most 0.006 from levels 2 to 3. Now, the R^2 improvements are at least 10x that number (0.06). We think these explanations also make more sense.
>
> >> “The contribution seems incremental, given that Tsang et al (2018) already explored explanations based on interactions.”
>
> Thanks for acknowledging our contribution of context-free explanations. Although applying NID on locally sampled points might seem simple for context-dependent explanations, there are several reasons why we were compelled to report this approach and experiments for it. We believe that broadening the application of NID to hierarchical explanations is important for the following reasons (in addition to the contributions noted in our original paper submission):
>
> 1)	Quantification: The quantification of explanations provided by interpretability methods has become increasingly important (Kim et al. 2018, Bau et al. 2017). Given that there have been several works recently on local interaction interpretations (Murdoch et al. 2018, Lundberg et al. 2018, Greenside et al. 2018), we hope that our work sets additional guidelines for local interaction evaluation, using both existing evaluation methods and a new one. In particular, we promote the following quantifications: evaluation of interaction explanations w.r.t synthetic ground truth (Section 5.2.1), approximation evaluation (Table 3 and R^2 scores in 5.2.3), and the novel context-free consistency (Section 5.3), which assumes that the local interaction explanations work. Regarding context-free consistency, we have added comparisons between Mahe and baselines: GA2M (Lou et al. 2013) and GLM with pairwise interactions, demonstrating that Mahe's interaction attributions tend to be more consistent than the baselines' before and after interaction negation (Appendix D).
>
> 2)	Rich interactions in general deep learning: In this work, we have demonstrated that different forms of deep learning models are rich with context-dependent (and -free) interactions in a variety of domains (e.g. Table 3). This is made possible by applying interaction detection beyond the conventional tabular datasets studied in Tsang et al. (2018) and other works. Please note that the linear approximation of univariate GAMs does not affect Mahe’s approximation performance in most of our results from Table 3 (via our response to Reviewer 3, point 3).
>
> 3)	High-order interaction explanations: For the interaction detection literature and the models that fit interactions (e.g. GAMs in Eq. 3), visualizing high-order interactions has been challenging (Tsang et al. 2018). For example, while a GA2M can be used to visualize pairwise interactions in 3D (Lou et al. 2013), visualizing higher-order interactions the same way would require understanding high-dimensional space (i.e. 4D and higher). We believe that explaining high-order interactions locally via interaction detection and GAMs is a compelling way to visualize complex interacting behavior, especially in black-box models (e.g. high-order interactions in Section 5.2.3; the CACGTG 6-way interaction in Section 5.3, which can also be visualized).
>
> Besides these context-dependent contributions, the methodological contribution of finding context-free explanations can also stand as a new evaluation metric for future research on speeding up or learning context-free interactions.
>
> Out of these contributions, we think that the most important are written in the paper.
>
>
> References:
>
> David Bau, Bolei Zhou, Aditya Khosla, Aude Oliva, and Antonio Torralba. Network dissection: Quantifying interpretability of deep visual representations. In CVPR, 2017.
> Peyton Greenside, Tyler Shimko, Polly Fordyce, and Anshul Kundaje. Discovering epistatic feature interactions from neural network models of regulatory dna sequences. Bioinformatics, 2018.
> Been Kim, Martin Wattenberg, Justin Gilmer, Carrie Cai, James Wexler, Fernanda Viegas, et al. Interpretability beyond feature attribution: Quantitative testing with concept activation vectors (tcav). In ICML, 2018.
> Yin Lou, Rich Caruana, Johannes Gehrke, and Giles Hooker. Accurate intelligible models with pairwise interactions. In KDD, 2013.
> Scott M Lundberg, Gabriel G Erion, and Su-In Lee. Consistent individualized feature attribution for tree ensembles. arXiv preprint, 2018.
> W James Murdoch, Peter J Liu, and Bin Yu. Beyond word importance: Contextual decomposition to extract interactions from lstms. In ICLR, 2018.
> Michael Tsang, Dehua Cheng, Yan Liu. Detecting Statistical Interactions from Neural Network Weights. In ICLR, 2018.

---

### Official Review · AnonReviewer3 · 2018-11-05
**interesting paper with a thorough evaluation**

**Rating:** 6
**Confidence:** 4

**Review:**

Summary

This paper proposes a method named Mahe that can provide hierarchical explanations for a model: including both context-dependent(instance level) and context-free (global) explanations by a local interpretation algorithm. It obtains context-free explanations through generalizing context-dependent interactions to explain global behaviors. The effectiveness is shown through a number of synthetic and real-world data experiments.

The paper provides an interesting way to get context-free explanations from local explanations. The experiments are well designed and the paper is overall written well.

Major comments

0. The motivation of the local-MLP models is not convincing.

1. Of particular concern is the computational time cost of the model, as it involves retraining and an exhaustive search through local interactions to get context-free explanations.

The paper provides no experiments about timing cost to show the relative computational scalability of the proposed method. As Mahe trains MLPs per data sample and searches through all interactions for finding context-free explanations, this raises concerns.

2. The paper includes no baseline comparisons for finding context-free interactions.

3. Non-linear GAM is replaced by linear approximations in the experiments. More experiments showing the advantage of non-linear function approximation is recommended.

4. Minor Comments: In the description, "L + 1 different levels of a hierarchical explanation which constitutes the context-dependent explanation", What does L indicate? The order of interactions?

---

> ### Author Response · Authors · 2018-11-24
> **Author Response**
>
> Dear reviewer,
>
> Thank you for your suggestions and support for our paper. We have conducted the major experiments you suggested, and our responses to your comments are below:
>
> >> “0. The motivation of the local-MLP models is not convincing”
>
> Thank you for your comment. Our motivations for using local MLP models are threefold: 1) MLPs are universal function approximators, which is an especially important property for learning arbitrarily complex, i.e. non-additive, interactions. 2) MLPs were able to obtain state-of-art level accuracy and efficiency at interaction detection, which is why we use the MLP with NID. The advantages of 1) and 2) have been clarified in Section 4.1 in the last paragraph of “Local Interaction Detection”. The third motivation for using local MLPs comes from our experiments on your second point. After performing experiments comparing Mahe to baselines for identifying context-free interactions, we found that local MLPs in the form of the GAM with interactions (Eq. 3) outperformed the baseline tree-based approach (Lou et al. 2013) that was also designed to learn the same form of function (GAM with interactions).
>
> >> “1. Of particular concern is the computational time cost of the model, as it involves retraining and an exhaustive search through local interactions to get context-free explanations… The paper provides no experiments about timing cost…”
>
> We agree that using an exhaustive search is a concern for runtime, which is why we originally included the part of Context Free Explanations (Section 4.2) to locally modify the target model to check if gives a global response. The local modification takes of role of checking whether any globally consistent behavior is more than just coincidence, which can be useful when the contexts available tend to biased in one way or another (for example, all contexts having the overall same prediction polarity as the interaction’s polarity).
> Nevertheless, individual testing on data instances is indeed needed in Mahe. We have added runtime experiments evaluating the average time needed for context-dependent (Figure 10), and -free (Figure 11) explanations for explaining the models trained on real-world datasets. From context-free runtime experiments, we found that sequentially testing over 40 instances before and after model modification (80 instances total) often takes less than two hours for a single interaction. If one has extra computational resources are available, individual tests can be parallelized.
>
> >> “2. The paper includes no baseline comparisons for context-free interactions”
>
> Thank you for this comment. We have performed experiments comparing Mahe to baselines for identifying context-free interactions. Using the same context-free interactions identified from Sentiment-LSTM in Figure 5, we evaluate whether baseline methods for interaction detection and fitting can also identify them. The baselines we use are lasso-regularized GLM with all pairs of multiplicative interactions (Bien et al. 2013) and GA2M (Lou et al. 2013), a nonlinear tree-based model for fitting pairwise interactions in Eq. 3. Experimental results are shown in Appendix D. We found that while both baselines were originally close to performing as well as Mahe in identifying consistent interaction polarity, the baselines – especially GLM – did not perform nearly as well after using them to locally modify Sentiment-LSTM to identify consistent negated behavior.
>
> >> “3. Non-linear GAM is replaced by linear approximations in the experiments. More experiments showing the advantage of non-linear function approximation is recommended”
>
> For our experiments explaining most models trained on real-world datasets, replacing linear approximations with nonlinear (univariate) GAM made no difference because our feature representations here were always binary, and a linear approximation can perfectly fit two points. However, we noticed significant performance improvements when the task was binary classification rather than regression, which was the case for explaining Transformer (from 0.75 to 0.95 avg AUC corresponding to the experiment in Table 3, K=0). We believe there was performance improvement because binary classification can have multiple solutions, and a better solution was found by the more flexible GAM.  Regarding the advantages of nonlinear approximation of interactions, we believe our results for point 2 are relevant.
>
> >> “4. Minor Comments”
>
> L indicates the number of hierarchical levels with interactions. This has been clarified in the relevant section in Section 4.1.
>
>
> References:
>
> Jacob Bien, Jonathan Taylor, and Robert Tibshirani. A lasso for hierarchical interactions. Annals of statistics, 2013.
> Yin Lou, Rich Caruana, Johannes Gehrke, and Giles Hooker. Accurate intelligible models with pairwise interactions. In KDD, 2013.

---

### Meta-Review · Area_Chair1 · 2018-12-17
**Expensive approach, unclear writing**

**Confidence:** 3
**Recommendation:** Reject

**Metareview:**

This paper introduces Mahe, a model-agnostic hierarchical explanation technique, that constructs a hierarchy of explanations, from local, context-dependent ones (like LIME) to global, context-free ones. The reviewers found the proposed work to be a quite interesting application of the neural interaction detection (NID) framework, and overall found the results to be quite extensive and promising.

The reviewers and the AC note the following as the primary concerns of the paper: (1) a crucial concern with the proposed work is the clarity of writing in the paper, and (2) the proposed work is quite expensive, computationally, as the exhaustive search is needed over local interactions.

The reviewers appreciated the detailed comments and the revision, and felt the revised the manuscript was much improved by the additional editing, details in the papers, and the additional experiments. However, both reviewer 1 and 3 have strong reservations about the computational complexity of the approach, and the additional experiments did not alleviate it. Further, reviewer 1 is still concerned about the clarity of the work, finding much of the proposed work to be unclear, and recommends further revisions.

Given these considerations, everyone felt that the idea is strong and most of the experiments are quite promising. However, without further editing and some efficiency strategies, it barely misses the bar of acceptance.